# Current Understanding of the Molecular Basis of Spices for the Development of Potential Antimicrobial Medicine

**DOI:** 10.3390/antibiotics12020270

**Published:** 2023-01-29

**Authors:** Purnima Khatri, Asha Rani, Saif Hameed, Subhash Chandra, Chung-Ming Chang, Ramendra Pati Pandey

**Affiliations:** 1Centre for Drug Design Discovery and Development (C4D), SRM University, Sonepat 131029, India; 2Department of Microbiology, SRM University, Sonepat 131029, India; 3Amity Institute of Biotechnology, Amity University Haryana, Gurugram 122413, India; 4Computational Biology & Biotechnology Laboratory, Department of Botany, SSJ Campus, Soban Singh Jeena University, Almora 263601, India; 5Master & Ph.D. Program in Biotechnology Industry, Chang Gung University, No. 259, Wenhua 1st Rd., Guishan Dist., Taoyuan City 33302, Taiwan

**Keywords:** phytochemical, antimicrobial resistance, spices, antifungal, antibacterial

## Abstract

Antimicrobial resistance increases day by day around the world. To overcome this situation new antimicrobial agents are needed. Spices such as clove, ginger, coriander, garlic, and turmeric have the potential to fight resistant microbes. Due to their therapeutic properties, medicinal herbs and spices have been utilized as herbal medicines since antiquity. They are important sources of organic antibacterial substances that are employed in treating infectious disorders caused by pathogens such as bacteria. The main focus of the study is the bioactivity of the active ingredients present in different kinds of naturally available spices. We conducted a thorough search of PubMed, Google Scholar, and Research Gate for this review. We have read many kinds of available literature, and in this paper, we conclude that many different kinds of naturally available spices perform some form of bioactivity. After reading several papers, we found that some spices have good antimicrobial and antifungal properties, which may help in controlling the emerging antimicrobial resistance and improving human health. Spices have many phytochemicals, which show good antimicrobial and antifungal effects. This review of the literature concludes that the natural bioactivate compounds present in spices can be used as a drug to overcome antimicrobial resistance in human beings.

## 1. Introduction

India is a country that is famous for its spices. Since ancient times, people have used spices in their kitchens to enhance the flavor of their food and for therapeutic purposes [1]. Many people use spices to treat microbial infections including bacterial fungal and viral ones. Some spices such as turmeric, garlic, asafoetida, clove, ginger, and many more are routinely used in India to cure diseases [2,3]. More than 500 thousand plants exist on Earth, but only 10% of plants are known to be used by humans and animals [1,4]. In the last few decades, the number of multidrug-resistant microbes has increased gradually [4]. Humans use too many antibiotics and antifungals, creating endurance in microbes, and thus, they are becoming resistant to them. Various microbial infections such as bacterial and fungal infections cause life-threatening diseases as a result of microbes such as *Escherichia coli*, *Klebsiella pneumoniae*, *Pseudomonas aeruginosa*, *Neisseria gonorrhea*, *Candida albicans*, and *Candida auris*, etc. [5], which are multidrug resistant and cause several diseases such as cholera, respiratory syndrome, bacterial meningitis, urinary tract infections, Candidiasis, and Aspergillosis [6]. The harrowing conditions caused by infectious diseases as a result of drug-resistant microbes and antibiotic side effects has directed our interest toward herbal and spice plants that are used as natural drugs and have fewer side effects than synthetic drugs do [7,8]. In recent years, many researchers have shown interest in spices for treating some infectious diseases to check the antimicrobial activity of spices against a wide range of bacteria, yeasts, and fungus [9,10]. Many spices such as coriander, cinnamon, gooseberry, turmeric, clove, fenugreek seed, asafoetida, star anise, garlic, black pepper, bay leaf, and curry leaf, etc., show good antimicrobial activity [11,12]. *Curcuma longa*, which is commonly called turmeric, is used in food, gives an orange-yellow color, and is very efficient against some fungus and bacterial strains. Its most active compound is curcumin, which has many benefits that have been scientifically proven by researchers [13,14]. In the literature review, the researchers advise that curcumin may be helpful for preventing metabolic syndrome and inflammatory conditions, and it also helps to improve heart disorders [15,16]. Indian gooseberry’s scientific name is *Emblica officinalis*, but it is commonly called “Amla” in India [17]. It adds a sour flavor to food. It is used to cure several skin disorders, and also, to recover immunity [18]. Its ethanol and methanol extracts show good activity against both bacterial and fungal strains [1]. *Foeniculum vulgare* essential oils work very well against many bacterial and fungal strains in India, and they are commonly called “Saunf” or fennel seeds [19]. Fennel seed has a sweet smell, and is highly nutritious. Chewing fennel seed help with digestion, and it may prevent many abdominal-related diseases [20]. “Hing” is a spice that has a strong odor, and its scientific name is *Ferula assafoetida* [21]. It has been used to cure coughs, asthma, stomach pain, blood pressure, and menstrual cramps, etc. Spices and their ingredients are generally recognized as safe (GRAS) [22], and their use is permitted by various regulatory agencies such as the (FDA), the European Union standards, and the Food safety and standard authority of India (FSSAI) [23]. Overall, spice extracts and their essential oils have the potential to prevent microbial infections [24]. Therefore, the extract of the spices is used as a natural drug to inhibit bacterial and fungal infections [25]. The main active compound of essential oil is terpenes, and they are phenolic and have antimicrobial properties [10]. All active phenolic compounds have a different target to eliminate harmful pathogens. A rapidly increasing number of cases of multidrug-resistant pathogens are causing a big problem in society. To minimize this problem, we need some new drugs which are highly efficient and have fewer side effects [26]. In this review of the literature, we summarize several important spices and their antimicrobial properties to propose new drugs that are less harmful and more effective. The essential oils of some spices are efficient at preventing harmful diseases.

## 2. Medicinal Value of Spices

All spices have different properties, and they are used for many purposes. We all know that mainly spices are used in the kitchen to enhance the flavor of our food, but some spices also have therapeutic properties [27]. In ancient periods, spices were used as medicine in India, and these spices have an extensive variety of biological and pharmacologic properties. Due to their high phenolic component content, spices including cinnamon, clove, rosemary, and oregano are great sources of antioxidants [28,29]. The exact function of spices and herbs in preserving health, particularly in preventing the emergence of chronic, non-communicable diseases, is currently unknown. In this overview, common spices such as garlic, gooseberry, fenugreek, black pepper, turmeric, chili pepper, and ginger, and others are highlighted for their possible health advantages [6]. In Table 1, the medicinal usage of spices and their active compounds are given below.

## 3. Antimicrobial Property of Spices

Several research articles suggest that spices such as curcumin, clove, oregano, thyme, star anise, cumin, black pepper, garlic, coriander, and many more spices have antimicrobial properties, and they are used to treat antibacterial and antifungal infections [6,7,27]. The activity of spices depends on their extraction method and which solvent was used for their extract [25,26,36]. Numerous spices produce essential oils, which are more effective against different types of bacterial and fungal strains [33,34]. Essential oils, which are also called volatile oils when they are extracted from plants containing hydrophobic liquid volatile compounds, are widely used in aromatherapy and several therapeutic systems [25]. As a result, those in the scientific sector are looking at them even more for the treatment of various illnesses, such as cancer, HIV, and asthma [23].

### 3.1. Cloves

In order to enhance the flavor and scent of our food, *Syzygium aromaticum*, a member of the Myrtaceae family, is frequently used as a spice [36]. Eugenol is a phytochemical that is extracted from cloves and used to alleviate toothache pain and other types of pain. Several research projects on cloves are checking their efficacy against many pathogens [6]. Both clove essential oil (EO) and eugenol are phenolic molecules that can deactivate peptides, alter the composition of cellular membranes, and prevent the growth of some varieties of yeast and Gram-positive and Gram-negative bacteria [37]. Due to their medicinal properties, they are used as antiseptics in dentistry to overcome dental pain [38].

#### 3.1.1. Phytochemical of Cloves

Many investigations have been conducted to identify different *S. aromaticum* components [39]. Between fifteen and twenty percent of the essential oils found in clove buds are made up of eugenol, eugenyl acetate, and β-caryophyllene [38]. Clove oil also contains triterpenoids, including alpha-linolenic acid, vanillin, categoric acid, tannins, gallotannic acid, methyl salicylate, eugenin, kaempferol, rhamnetin, and eugenitin [6]. The distinctively pleasant aroma of cloves is caused by minor ingredients such as methyl amyl ketone, and methyl salicylate, etc. [40]. Some phytochemical properties of clove show in Figure 1.

#### 3.1.2. Antimicrobial Activity of Cloves

Numerous studies have shown that cloves have strong antibacterial properties. Many compounds, including eugenol, iso-eugenol, methyl-eugenol, phenyl propanoides, dehydro-dieugenol, and trans-confireryl aldehyde, are present [42]. These substances have the ability to denature proteins and interact with lipids in the cellular membrane to influence their porosity. Due to the lysis of the spores and micelles, eugenol was discovered during a chromatographic analysis to be the primary component responsible for antifungal activities. Devi et al. [43] similarly observed a similar mode of action for membrane ruptures and macromolecule deformations caused by eugenol. For *Candida*, *Aspergillus*, and *dermatophytes*, a wide range of fungicidal activities were documented, and their mechanisms of action were related to perforations of the cell membranes [44].

### 3.2. Cinnamon

The botanical name for cinnamon is *Cinnamomum verum*, and it is an evergreen plant, and the spices are obtained from its bark [45]. The spice is brown in color, and it has a pleasant fragrance and a sweet flavor. Its essential oil is used in drugs, perfume, flavoring, and liqueur. The major phytochemical compound present in their essential oil is cinnamaldehyde, which is responsible for their antimicrobial property [46]. Cinnamon extract has various therapeutic uses; it shows antifungal, antibacterial, anti-diarrheal, antisemitic, and insecticidal properties [47]. Cinnamon is also used in aromatherapy, which is the medicinal application of plant essential oils that enter the body through the skin or the nose [9]. There are several types of research on cinnamon spices regarding their different roles in the medicinal field [7].

#### 3.2.1. *Phytochemicals of Cinnamon*

Cinnamon is made up of many resinous substances, such as cinnamaldehyde, cinnamate, and cinnamic acid, and a large number of essential oils [48]. According to Singh et al. [49], the spicy flavor and scent are caused by the presence of cinnamaldehyde, and they occur as a result of oxygen absorption. In Figure 2 we represent the structure of cinnmaldehyde. The hue of cinnamon darkens with age, enhancing the resinous components. Sangal reported that cinnamon has a number of physiochemical qualities [49].“Trans-cinnamaldehyde, cinnamyl acetate, eugenol, L-borneol, caryophyllene oxide, b-caryophyllene, L-bornyl acetate, E-nerolidol, α-cubebin, α-terpineol, α-terpinolene, and α-thujene are just a few of the essential oils that have been found” [50].

#### 3.2.2. *Antimicrobial Activity of Cinnamon*

Numerous antibacterial properties of cinnamon and its oils have been documented to date in numerous research studies [48]. For example, Matan et al. stated the effects of cinnamon oils on various bacterial, fungal, and yeast species, which indicate cinnamon as a natural antimicrobial agent [51]. According to Goi et al., combinations of cinnamon and clove oils have antibacterial properties both against Gram-positive and Gram-negative bacteria, including *Listeria monocytogenes*, *Enterococcus faecalis*, *Staphylococcus aureus*, and *Bacillus cereus* [52]. According to a study by Hili et al., cinnamon oils may be effective against yeast and a variety of bacteria, including *Pseudomonas aeruginosa*, *Staphylococcus aureus*, and *Escherichia coli* [53]. A recent study described the effectiveness of cinnamon and other plant extracts against oral bacteria. Overall, compared to other examined plant extracts such as *Azadirachta indica* and *Syzygium aromaticum*, cinnamon’s essential oil is more potent [54].

### 3.3. Cardamom

Green cardamom’s botanical name is *Elettaria cardamomum*, and it fits into the Zingiberaceae family [55]. It is used as a flavoring additive in food preparation, drinks, and medication [56]. Cardamom shows good antibacterial and antifungal activities Many research studies suggest that cardamom helps to boost the immune system, reduce high blood pressure, reduce long-term inflammation, cure digestive problems, and also, improve breathing [57]. Researchers on this spice are developing new antibiotics that are less harmful to health.

#### 3.3.1. Phytochemicals of Cardamom

Depending on the species, plant sections, and extraction techniques used, the yield of the EO from dry cardamom ranged from 0.2% to 8.7% [58]. It contains 1,8-cineole, α-terpinyl acetate, α-terpineol, sabinene, nerol and α-pinene α-terpinyl acetate [55], α-terpineol 1, 8-cineole, sabinene, linalyl acetate, linalool limonene [59], 4-terpineol [60], geranylacetate, cis-sabinene hydrate acetate, β-caryophyllene, β-selinene, γ-cadinene, translinalooloxide [61], Geraniol, 1,8-Cineole, β-terpineol, 4-terpineol, and 1,8-Cineol, α-terpene [62]. Main active compound is cineole and α- terpinol shown in Figure 3.

#### 3.3.2. Antimicrobial Activity of Cardamom

Cardamom essential oil (CEO) has potent antibacterial properties against a range of food-borne pathogens. Cardamom oil application moderately inhibited the growth of *Morgenella morganii* [55]. CEO (10 mg/mL) showed antibacterial activity against *S. aureus*, *E. coli*, and *C. albicans* [63]. Therefore, CEO could be extremely important in creating new and safe antibiotics for use in modern medicine [64]. CEOs may be used to prevent damage from food-borne illnesses and microbial agents that cause food to rot because of their potential for diverse antibacterial and antifungal activities [65,66]. The disc diffusion approach has been used in the bulk of investigations looking at the antibacterial activity of cardamom extracts and CEO, however, because of its flaws, it must be combined with the more useful MIC test [67].

### 3.4. Coriander

*Coriandrum sativum* is widely used in spices all parts of this plant that have a significant role in their properties [68]. They contain several compounds such as thymol, bornyl acetate, gallic acid, and many more. The essential oil of coriander contains polyphenol and terpenes, the major constituent of coriander is linalool, which has several medicinal properties and strong flavors its structure given in Figure 4 [69]. Its seeds are consumed for relieving pain, inflammation, and rheumatoid arthritis, whereas its extraction is used for eye problems and mouth ulcers. Coriander shows very good activity against food-borne pathogens such as *Campylobacter* and *Salmonella*, and in addition, it is also used as an antioxidant, for indigestion, and diabetes [70].

#### 3.4.1. Phytochemical of Coriander

A medicinal plant called *Coriandrum sativum* L. is indigenous to the eastern Mediterranean region, from which it has spread to various parts of the world, along with many other aromatic species [72]. In this context, the primary and secondary metabolite of coriander is the essential oil. However, the current compilation also makes reference of an additional group of active ingredients.

Fruits contain carbohydrates, alkaloid, quercetin, resins, tannins, quinones, sterols, and fixed oils [73]. The parts of the coriander fruits that are thought to be most important are the essential oil and fatty oil. The fatty acids present in coriander fruits include palmitic acid, cis-6-octadecenoic acid, linoleic acid, and oleic acid [74]. As per the reports, coriander is a very good source of thiamine, zinc, and dietary fiber, and like all other green leafy vegetables, it has very low amounts of saturated fat and cholesterol, as well as a rich supply of vitamins, minerals, and iron [75]. Eighty-four percent of unripe coriander is water. Here, the order of the most significant phytoconstituents are described [12].

#### 3.4.2. Antimicrobial Activity of Coriander

The essential oil concentration of coriander is what gives it its antibacterial properties [76]. Additionally, it was discovered that coriander’s aqueous extract was effective against bacteria that cause acne (the MIC values for Propionibacterium acne and *Staphylococcus epidermidis* are within 1.7–2.1 mg/mL, respectively) [77]. The widely viable formulations for the treatment of acne displayed similar activities. Due to its antibacterial properties, coriander oil is a fantastic choice for the development of cutting-edge anti-acne compositions [78]. The antioxidant, anti-inflammatory, analgesic, and antibacterial qualities of coriander make it a valuable herbal therapy for diaper dermatitis, a common skin condition. Additionally, coriander oil has demonstrated potent activity with varying degrees of inhibition against *Bacillus cereus*, *Enterococcus faecalis*, *Staphylococcus aureus*, *Pseudomonas aeruginosa*, and *Acinetobacter baumannii* [79]. With the exception of *Bacillus cereus* and *E. faecalis*, *P. aeruginosa* was the most resistant strain to growth inhibition by the tested oil, displaying the highest determined MIC, along with one of the multidrug-resistant clinical isolates of *Acinetobacter baumannii* [79]. As a result, the use of coriander oil in antibacterial formulations can be encouraged because it efficiently eliminates the harmful bacteria linked to hospital infections and foodborne illnesses [80]. The antibacterial and antifungal properties of coriander essential oils have been examined in numerous research, and they have been determined to have good potency. The essential oil from *Coriandrum sativum* produced by hydrodistillation was tested against various fungi, and it showed antifungal action against *Candida*, with the exception of *C. tropicalis* CBS_94_ [81]. As a result, it was determined that the oil had potential as an antibacterial agent for treating or preventing *Candida* yeast infections. The fruit oil of coriander was found to exhibit excellent efficacy against *Listeria monocytogenes*, *S. aureus*, *S. haemolyticus*, *P. aeruginosa*, and *E. coli* [82]. Additionally, coriander essential oils extracted using both hydro distillation and microwave assistance have been tested for their antibacterial potency. Other than the time and energy savings, no noticeable differences in the activities were discovered [83].

### 3.5. Curry Leaf

The botanical name is *Murraya koenigii,* and it belongs to the Rutaceae family, and it is more important because of its traditional medicinal uses [84]. All of the parts of the curry plant show many biological activities; mainly, the leaves are used for their aroma and flavor. The main active compound is carbazole alkaloids, which have antibacterial, antioxidant, antidiabetic, anti-inflammatory, and anti-tumor properties [85]. Curry leaf extract shows antifungal and antibacterial activities against many strains. *M. koenigii* leaves contain alkaloids that have been investigated and found to have long-lasting anti-diabetic effects and inhibit the aldose reductase enzyme, glucose consumption, and other enzyme systems [86]. To learn about their efficacy, more experimental studies should be conducted.

#### 3.5.1. Phytochemical of Curry Leaf

The leaves, roots, and stem bark of *M. koenigii* have been used to isolate a wide variety of phytochemicals. Alkaloids, flavonoids, terpenoids, and polyphenols have been produced by the *M. koenigii* extracts of its leaves, roots, stem bark, fruits, and seeds [87]. The leaves have been shown to contain considerable amounts of vitamins, including vitamin A (B-carotene), vitamin B3, which is sometimes known as niacin, and vitamin B1 (thiamin) [88]. Linallol is main active compounds its chemical structure show in Figure 5. The proportion of the alcohol-soluble extract is 1.82%, that of the ash is 13.06%, that of the acid-insoluble ash is 1.35%, that of the extractive in cold water (20 °C) is 27.33%, and that of the maximum extractive in hot water is 33.45%. Numerous terpenoids, flavonoids, essential oils, and carbazole alkaloids have positive effects [89].

#### 3.5.2. Antimicrobial Activity of Curry Leaf

Several investigations have reported *M. koenigii’s* antifungal activity. For instance, it has been claimed that the leaves’ essential oil has antifungal properties [90]. The presence of phytochemical components with complex molecular structures and a variety of action mechanisms, such as alkaloids, terpenoids, flavonoids, phenolics, tannins, and saponins, are known for their antimicrobial properties [91]. The active ingredients of *M. koenigii* have a notable capacity for inhibiting mycelial growth, which supports antifungal efficacy. Unplanned antibiotic use encourages the growth of numerous dangerous, drug-resistant pathogenic bacterial strains, and there are not enough effective cures for these diseases [92]. There is still a need to find novel antimicrobials. Currently, the interest in alternative medications, such as natural or herbal remedies, in addition to antibiotics and chemically synthesized drugs, is rising. Due to their natural sources, they can have fewer side effects or be less toxic [84]. It is often difficult to treat microbial infections without experiencing negative effects. In this context, the search for strong compounds from natural herbal remedies is ongoing, in addition to the use of synthetic medications and conventional antibiotics [93]. Extracts from *M. koenigii* have been shown to have antibacterial activities against a range of microbes [93]. Therefore, *M. koenigii* leaves could be utilized effectively as a natural medicine in the consumption of regular meals to avoid a variety of bacterial diseases. *Staphylococcus aureus* and *Klebsiella pneumonia* bacterial strains are resistant to the antibacterial effects of pyranocarbazoles obtained from *M. koenigii* [94]. Sustainably generated *M. koenigii* silver nanoparticles (AGNPs) demonstrated therapeutic effectiveness against MDR bacteria [93]. *Pseudomonas aeruginosa* was resistant to the antibiofilm effects of *M. koenigii* essential oil, and it was observed that a treatment with the oil resulted in an 80% decrease in the rate of *P. aeruginosa* biofilm development. Microscopic examinations have supported the finding that the *M. koenigii* essential oil treatment reduced *Pseudomonas aeruginosa’s* ability to produce biofilms [95].

### 3.6. Gooseberry

*Phyllanthus emblica* is a fruity spice powder, and it gives sweet sour and pungent flavor to food [96]. Its fruits added various nutritious elements to our food: vitamin C, minerals, nicotinic acid, methionine, tryptophan, and phosphorous, and many more nutraceuticals [17]. Indian gooseberry is traditionally used in Please ensure that meaning is retained. to boost the immune system [96]. It is used to cure cancer, dental caries, obesity, tuberculosis, skin diseases, hyperacidity, hypertension, and other infectious diseases [97]. Numerous flavanols, which have been connected to advantages such as a better memory, are also present in gooseberries. It contains a soluble fiber that dissolves quickly in the body, which slows down the absorption of sugar. This may lessen blood sugar peaks [98]. These berries are also beneficial for the blood glucose and cholesterol levels found in patients with type 2 diabetes. The Indian gooseberry fiber aids in the body’s regulation of its bowel motions and may help with the symptoms of illnesses such as irritable bowel syndrome [1]. These berries contain high quantities of vitamin C, which aids your body in absorbing other nutrients. As a result, it might be beneficial if you take iron and other mineral supplements.

#### 3.6.1. Phytochemical of Gooseberry

The EO of gooseberry plants that has been examined the most is thought to contain a variety of phytochemicals, including alkaloids, benzenoids, flavonoids, terpenes, polysaccharides, and sterols, among others [99]. It should not come as a surprise that EO extracts have been linked to a variety of health-promoting characteristics. The benzenoids, which include organic acids, gallates, hydrolyzable tannins, and ellagitannins, make up the largest group of fruit’s active substances. the chemicals found in EO fruit that have been documented. Tannins, polyphenolic complex 1,3,6-trigalloylglucose, terehebin, corilagin, phyllantine, β-sitosterol, linoleic acid, ellagic acid, lupeol, and ascorbic acid are the major components of essential oils [100]. It has been proven that the ascorbic acid content of EO fruits is overestimated by the traditional spectrophotometric techniques (DNPH and indophenol-xylene). The enzymatic technique determined that fresh fruits had a vitamin C level of 213.539 mg/100 g [101]. Mucic acid lactone and ascorbic acid may be connected biosynthetically in EO fruits [102]. A fixed oil, phosphatides, and a minor amount of an essential oil with a distinctive scent are all present in the seeds [103]. Researchers have discovered that the 2,3-cis-configured tannins in EO bark are a combination of 3-O-gallated prodelphinidin and procyanidin [99].

#### 3.6.2. Antimicrobial Activity of Gooseberry

In the past few decades, a number of research studies have been carried out to evaluate the microbiological activity of various plant extracts. Several active components in the majority of the plant extracts were shown to have a strong antibacterial effect [17]. *Emblica officinalis* has demonstrated value in the design of efficient medicines and potent antibacterial characteristics [104]. Due to the antimicrobial properties of the EO extracts, which include antifungal, antiviral, and antibacterial activities, in biomedical research, this plant is being studied for the creation of unique and alternative antibacterial and complementary therapeutic options [96]. In Figure 6 we have mentioned many phytochemical properties of gooseberry. Gram-positive bacteria are more susceptible to the antibacterial effects of EO, although its ability to combat fungus is limited. The EO extracts when tested positive for *Candida albicans*, *Escherichia coli*, *Klebsiella pneumoniae*, *Staphylococcus aureus*, and *Bacillus cereus*, and they showed a large zone of inhibition (ZOI) [105]. The fruit of *P. emblica* has been used as a treatment for several microbial infections because of its antimicrobial activity against Gram-positive, Gram-negative, and fungal agents [106].

### 3.7. Garlic

*Allium sativum* belongs to the Alliaceae family, which is commonly called garlic, and its mostly used plant part is the bulb. The bulb structure is divided into numerous sections called cloves [107]. It is used as an antiseptic, stimulant, anthelminthic, diuretic, antisorbutic, and antiasthmatic, and for other purposes. According to recent studies, the use of garlic as a plant repellent against several plant pests and diseases is also advantageous [108]. Allicin is a molecule that gives a pungent smell to garlic, and it may have some pharmacological properties. Allicin may help to lower the blood pressure and maintain it within a healthy range, according to the research [109]. Although there are many traditional uses for garlic, there does not seem to be a lot which has been written about its insecticidal and antibacterial qualities, thus further research is necessary in this area [108].

#### 3.7.1. Phytochemical of Garlic

According to the claims, *A. sativum* bulbs contain hundreds of phytochemicals, including those that contain sulfur [108] such ajoenes, thiosulfinates (allicin), and sulfides (diallyl disulfide), etc. Alliin, the main cysteine sulfoxide, is converted into allicin by the enzyme allinase once the garlic is cut off and the parenchyma is dissolved [110]. The primary aroma compounds in freshly milled garlic homogenates are S-propyl-cysteine-sulfoxide (PCSO), allicin, and S-methyl cysteine-sulfoxide (MCSO) [111]. The pharmacological impact of allicin (allyl thiosulfinate), a sulfenic acid thioester, is due to both its interaction with thiol-containing proteins and its antioxidant activity [112]. Cysteine is converted into alliin during the production of allicin, which the allinase enzyme then hydrolyses. At room temperature, where two molecules are joined to make allicin, this enzyme divides the alliin into ammonium, pyruvate, and allyl sulfenic acid, all of which are extremely reactive and unstable substances [113].

#### 3.7.2. Antimicrobial Activity of Garlic

Garlic’s antibacterial qualities are assumed to be a result of its active component, allicin, which has been proven to be effective against a variety of pathogens, including antibiotic-resistant, Gram-positive, and Gram-negative bacteria such as *Shigella*, *Escherichia coli* [114], *Staphylococcus aureus*, *Pseudomonas aeruginosa*, *Streptococcus mutans*, and *S. faecalis* [115]. A number of pathogenic bacteria have been reported to be inhibited by different garlic extracts, including aqueous, chloroform, methanolic, and ethanolic extracts, with variable degrees of susceptibility [116]. For instance, ethanolic garlic extract has demonstrated a greater level of inhibitory activity against *E. coli* and *Salmonella typhi* than aqueous garlic extract did, which exhibited some to no inhibitory activities [116]. A variety of fungi, including species of *Candida*, *Torulopsis*, *Trichophyton*, *Cryptococcus*, *Aspergillus*, *Trichosporon*, and *Rhodotorula*, were successfully eradicated by garlic extracts [117]. In Figure 7 many phytochemicals property of garlic are mentioned. The germination and growth of *Rhodotorula mucilaginosa* and *Meyerozyma guilliermondii* have recently been discovered to be inhibited by the garlic extract [117]. The fungal cell walls were impacted by the garlic extract, which also resulted in irreversible changes to the fungal cells’ ultrastructure, which reduced their structural integrity and hindered their capacity to germinate. These cytoplasmic content changes result in damage to the cell organelles and nucleus, which ultimately causes cell death [115]. These substances work by penetrating the cell membranes and the organelle membranes, such as the mitochondria, and causing the breakdown of organelles and cell death [118].

### 3.8. Ginger

Ginger’s botanical name is *Zingiber officinale*, which belongs to the Zingiberaceae family, whose rhizome and ginger roots are used as a spice and also for medicinal purposes [119]. 

Approximately 400 compounds are present in ginger, and some volatiles oils are responsible for their aroma and flavor. Various compounds of volatile oils such as borneol, camphene, and linalool have many pharmacological properties such as immunomodulatory, anti-tumorigenic, anti-inflammatory, and anti-emetic ones [120]. Zingibain, a cysteine protease enzyme with characteristics that are similar to those of rennet, is another enzyme that has been found in fresh ginger [121]. Various research studies have proved that ginger essential oil performs an antimicrobial activity against different bacterial and antifungal strains [122]. There is much more to learn about the precise mechanisms of ginger’s antibacterial and antifungal effects. Phytochemicals of Ginger.

The prevalent active ingredients in ginger include phenolic and terpene chemicals [123]. Mostly gingerols, shogaols, and paradols make up ginger’s phenolic components. In Figure 8 structure of gingerols is shown. The primary polyphenols in fresh ginger are gingerols, including 6-, 8-, and 10-gingerol. Gingerols can be changed into matching shogaols through a heat treatment or extended storage. Shogaols can become paradols through hydrogenation [124]. In addition, ginger contains a variety of other phenolic substances, including quercetin, zingerone, gingerenone-A, and 6-dehydrogingerdione [125]. Additionally, ginger contains several terpene elements, including β-bisabolene, α-curcumin, and zingiberene, which are the primary components of ginger essential oils [35]. In addition to these, ginger also contains polysaccharides, lipids, organic acids, and raw fibers [35].

#### Antimicrobial Activity of Ginger

Due to antimicrobial resistance, the spread of pathogenic bacterial, fungal, and viral illnesses has posed a serious risk to the general public [126]. Many herbs and spices have been transformed into organic, powerful antibacterial compounds that are effective against a variety of harmful microbes. Ginger has reportedly demonstrated antibacterial, antifungal, and antiviral properties recently [127]. The development of biofilms has a significant role in both infection and antibiotic resistance. According to one study, ginger reduced the membrane integrity and prevented the generation of biofilms in a strain of *Pseudomonas aeruginosa* that was multidrug resistant [128]. Additionally, ginger’s crude extract and methanolic fraction reduced the virulence of *Streptococcus mutans* by inhibiting the production of glucan, biofilm formation, and adhesion [129]. A treated group of rats showed a decrease in the development of caries brought on by *Streptococcus mutans*, which is consistent with an in vitro investigation [129]. Additionally, an in vitro investigation found that gingerenone-A and 6-shogaol had an inhibitory impact on *Staphylococcus aureus* by preventing the action of the pathogen’s 6-hydroxymethyl-7, 8-dihydropterin pyrophosphokinase [130].

### 3.9. Turmeric

*Curcuma longa* belongs to the Zingiberaceae family, and it is commonly called “Haldi” in India. The rhizome of turmeric plants is also used as a spice, and it has a medicinal purpose [131]. Curcumin is an active compound of turmeric, and it is a very potent antioxidant, and it has potent anti-inflammatory properties, additionally, this organic compound has antiviral, antiviral, antiprotozoal, and antiparasitic activities [22]. In various studies on the antibacterial and antifungal activities of curcumin, there is a lack of information on how it affects diverse types of microorganisms, particularly clinical isolates and MDR strains [14]. Furthermore, this natural plant substance’s minimum inhibitory concentrations (MICs) against several prevalent human diseases in their planktonic forms have not yet been established [7]. The clinical studies have shown that curcumin supplementation has therapeutic advantages for people with cancer, including colorectal, pancreatic, and breast malignancies, as well as inflammatory illnesses [132]. Researchers are very interested in curcumin because of its diverse spectrum of biological activities and its pleiotropic medicinal potential.

#### 3.9.1. Phytochemical of Turmeric

Turmeric is composed of 5.1% fat, 6.3% protein, 69.4% carbs, 3.5% minerals [133], and 13.1% moisture. Sabinene (0.6%), borneol (0.5%), and a-phellandrene (1%), as well as cineol (1%), sesquiterpines (53%), zingiberene (25%), and curcumin (diferuloylmethane) (3–4%) are present in the essential oil made by steam distilling turmeric rhizomes [134]. There are volatile and nonvolatile chemicals in turmeric. Turmerone, zingiberene, curlone, and ar-turmerone are volatile chemicals [135]. Curcuminoids are among the nonvolatile ingredients. An oleoresin called turmeric has a heavy yellow-brown fraction and a light volatile oil component. It contains a significant amount of sesquiterpenoids, monoterpenoids, and curcuminoids. The flavonoid curcumin is one of turmeric’s primary active ingredients and their phytochemical properties shown in Figure 9 [13].

#### 3.9.2. Curcumin for Biogenic Nanoparticles Synthesis and Its Antimicrobial Potential

Biological organisms are used to create biogenic nanoparticles. The selection of biological bodies is made because they are simple to culture and have a high intracellular metal uptake rate. In addition to the extracellular secretion of enzymes, it provides straightforward downstream processing for product recovery with the convenience of handling biomass. The extracellular or intracellular manufacture of nanoparticles using various modes of synthesis is made possible by the metabolic activity of these microorganisms, resulting in the production of many types of biogenic nanoparticles. Depending on the cell and target backdrop, curcumin regulates different signaling molecules, enabling it to act on a variety of targets in the cellular pathways. However, because of its limited solubility in water and its crystalline structure, the bioactive component has a low bioavailability. To overcome these limitations, scientists have tried to boost the biological and pharmacological effectiveness of curcumin by making it smaller. Due to this, nanocurcumin was discovered, which boosts curcumin’s biological activity by enhancing its bioavailability, solubility, long-term circulation, and retention in the body [136]. Curcumin’s well-known antibacterial effect is regulated by its interaction with the FtsZ protein. In the majority of prokaryotic species, the FtsZ protein is in charge of a crucial stage of cell division. According to the reports, curcumin’s hydroxyl and methoxy groups in curcumin are closely related to its antibacterial effect. The FtsZ GTPase protein is catalyzed by the oxygen molecules of these functional groups attached to the phenolic rings of curcumin, leading to early cell death. Using *S. aureus* and *P. aeruginosa* as test subjects, Huang et al. showed that curcumin-encapsulated silver-polymeric NPs have antibacterial capabilities [137]. Similar to this, curcumin-capped micelles improved the antibacterial effects of miltefosine and alkyl phosphocholine erufosine against *S. aureus* [138]. Hepatitis C virus (HCV) cells treated with a nano complex had a reduced viral load, demonstrating the antiviral effects of curcumin micelles on HCV attachment. When the cells infected with the respiratory syncytial virus (RSV) were treated with curcumin-capped silver NPs, similar outcomes were seen [139]. 

#### 3.9.3. Antimicrobial Activity of Turmeric

According to a theory put forth by Odhav et al. [140], the mechanism underlying the antimicrobial effects of various spices includes the formation of hydrogen bonds with the membrane proteins and hydrophobic interactions with different phenolic compounds, which disturb the cell membrane, cause disruption to the cell wall, and harm the electron transport chain [141,142]. The anionic ingredients such as nitrate, chlorides, sulphates, and thiocyanate, as well as a number of other chemicals that are naturally found in plants, may be the cause of the aqueous extracts’ antibacterial activity. Since organic solvents dissolve the organic compounds quickly and release more potent antimicrobial components as a result, the ethanolic extracts have better benefits than the aqueous extracts do. In this instance, the thick structural components of Gram-positive bacteria may be to blame for the increased interaction between the structural lipoproteins, the active components, and curcumin. The Gram-positive bacteria may be inhibited as a result of the enhanced cooperation [141]. The inactivation of several cellular enzymes was connected to the antibacterial effect of various phenolic complexes, according to Moreno et al. [143]. This relationship depends on the rate at which the substances entered the cell and changes in the permeability of the membrane. The main component of a certain compound’s antibacterial effect is a change in cell membrane permeability. Phenolic substances have the potential to fully destroy the cell membranes, compromise cellular integrity, and ultimately, result in cell death [144].

As shown in Figure 10, Graphical representation of different extraction of spices using the data of Table 2.

## 4. Discussion

The extensive use of antibiotics causes resistance genes in bacteria, and the unrestricted repetition of antibiotics in the veterinary and medical fields is blamed for the emergence of such multidrug-resistant strains [26]. There is a demand for natural herbal alternatives to the routinely used antibacterial medicines due to the growth of multidrug-resistant bacteria. Public health has been extremely concerned about the foodborne illnesses brought on by eating contaminated food with both Gram-positive and Gram-negative bacteria [149]. Many thousands of years ago, plant essential oils and extracts were employed in food preservation, medicines, complementary medicine, and natural cures [150]. Therefore, in order to improve the standard of healthcare, it is essential to conduct scientific research on the plants that have traditionally been utilized in medicine [151]. Essential oils naturally occur in phytochemical compounds that have a variety of uses and have been used for treating many diseases all around the world for a very long time [152]. Natural goods such as essential oils and extracts may have less adverse side effects than synthetic pharmaceuticals of the same potency do, at least according to some data [153]. To create new antifungal drugs, which could encourage the use of the plant to treat a variety of infectious diseases, quantitative data on plant oils and extracts are required [6]. Additionally, there is a growth in the demand for effective, secure, natural remedies, as well as a revival of the interest in the organic treatment of human pathogenic fungal infections [33]. This review suggests that some spices show antimicrobial activities, which means that they can be used in future as a natural drug to overcome the antimicrobial resistance.

## Figures and Tables

**Figure 1 antibiotics-12-00270-f001:**
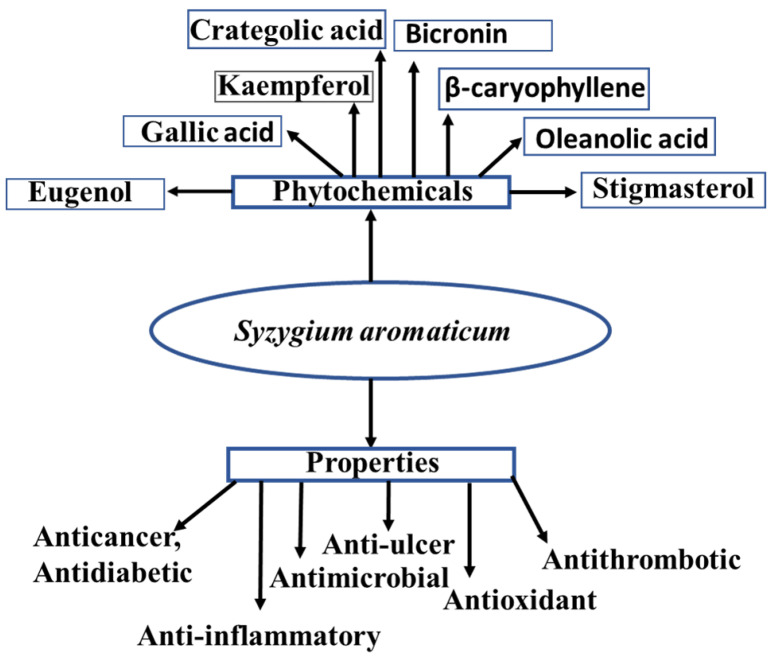
Phytochemical and therapeutic properties of cloves [41].

**Figure 2 antibiotics-12-00270-f002:**
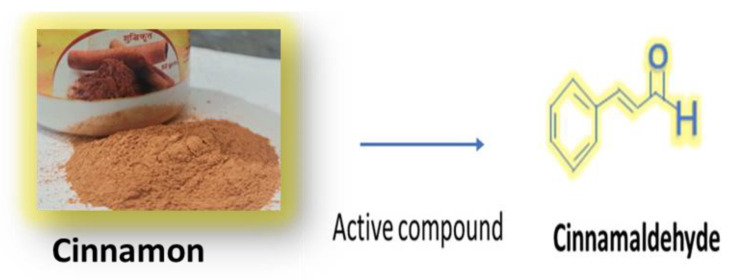
Active compound of cinnamon (Cinnamaldehyde).

**Figure 3 antibiotics-12-00270-f003:**
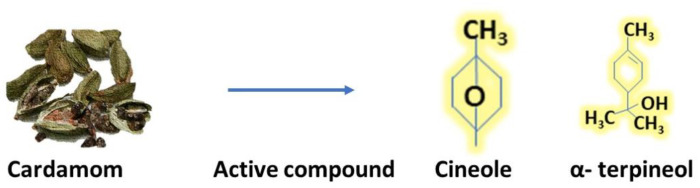
Active compound of cardamom (Cineole and α-terpineol).

**Figure 4 antibiotics-12-00270-f004:**
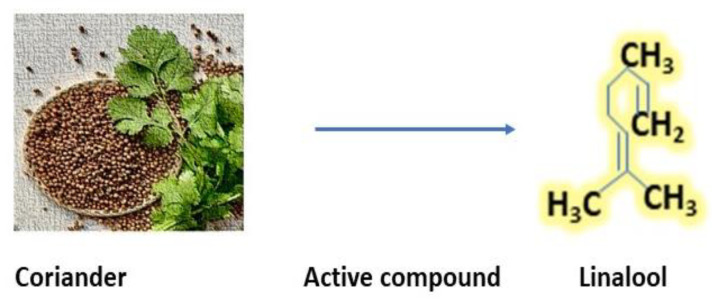
Active compound of coriander (Linalool) [71].

**Figure 5 antibiotics-12-00270-f005:**
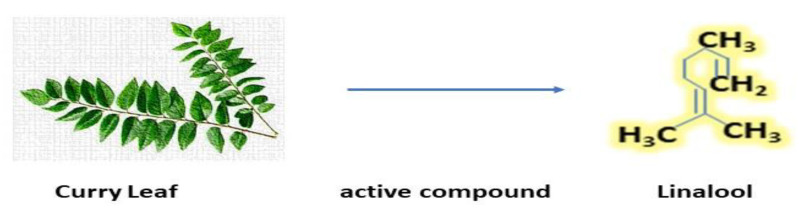
Most active compound of curry leaf (Linalool).

**Figure 6 antibiotics-12-00270-f006:**
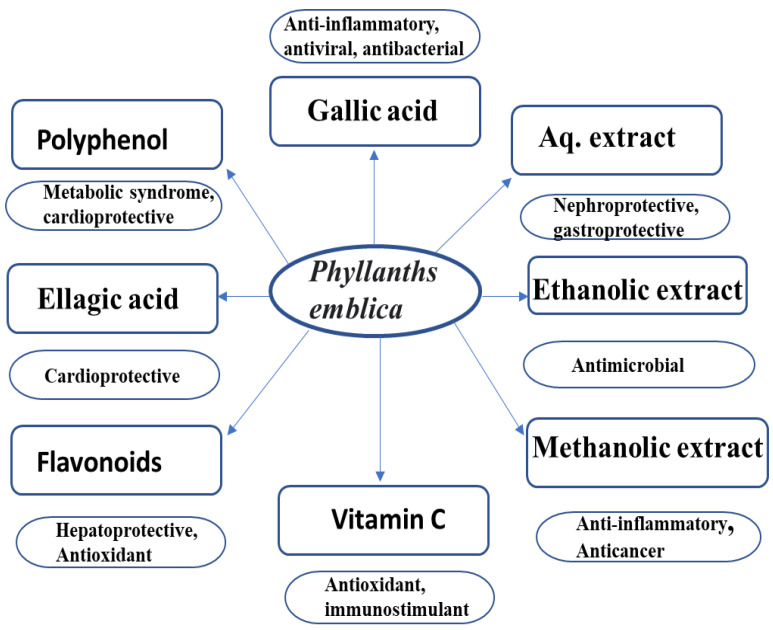
Phytochemicals and properties of gooseberry.

**Figure 7 antibiotics-12-00270-f007:**
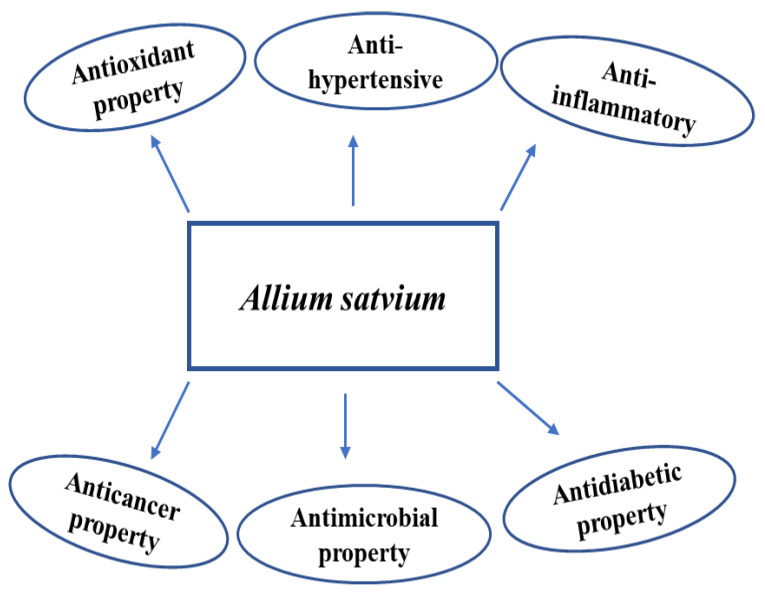
Pharmaceutical properties of garlic.

**Figure 8 antibiotics-12-00270-f008:**
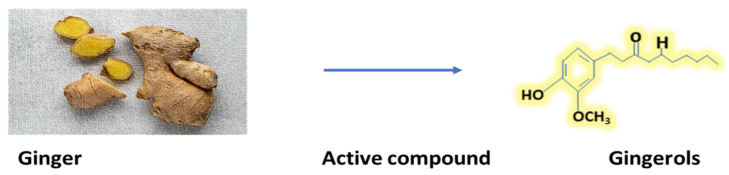
Most active compound of ginger (Gingerols).

**Figure 9 antibiotics-12-00270-f009:**
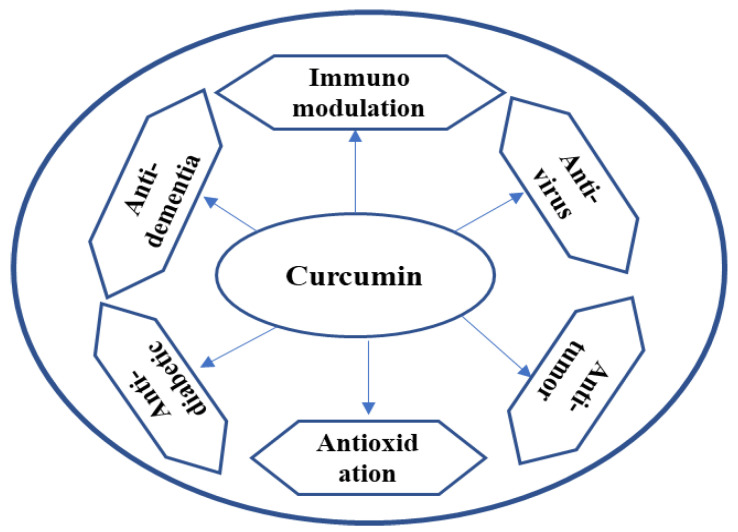
Curcumin main phytochemical of turmeric.

**Figure 10 antibiotics-12-00270-f010:**
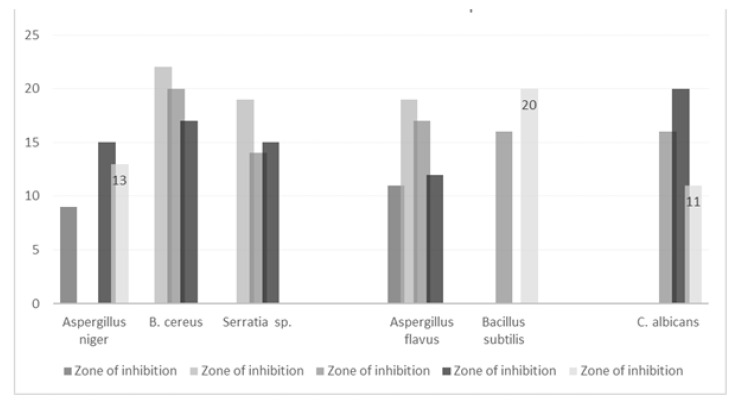
Zone of inhibition of microbial strain using different extraction of spices.

**Table 1 antibiotics-12-00270-t001:** Active compounds and medicinal uses of some spices.

Common Name	Botanical Name	Active Compound	Medicinal Uses and Benefits	References
Bay leaf	*Laurus nobilis*	Eugenol, methyl eugenol, and elemicin	Stimulant in sprains, narcotics, and in veterinary medicine.	[30]
Cumin	*Cuminum cyminum*	Aldehyde cumino	Parasiticidal, febrifuge, gastric, appetizing, used for skin diseases, flux, and canker.	[31]
Cardamom	*Elettaria cardamomum*	Cineole, pinene, sabinene, and porneol	Intoxicating, energizer, flatus relieving, peptic, used in cough mixture, and used in numerous therapeutic preparations.	[32]
Coriander	*Coriandrum sativum*	Geraniol	Carminative, diuretic, tonic, stimulant, stomachic, refrigerant, aphrodisiac, analgesic, and anti-inflammatory.	[32]
Clove	*Syzygium aromaticum*	Eugeniol	Sorbent, ocular, gastral, flatus relieving, restorative, spasmolytic, antibacterial, used in cough syrups, and rubefacient.	[23]
Garlic	*Allium sativum*	Allicin	Anti-cholesterol, antifungal, energizer, intoxicant, thermogenic, and used for coughs and asthma.	[33]
Ginger	*Zingiber officinale*	Gingerol and shogaol	Gastral, flatus relieving, placatory, foretaste, stomachal, rubefacient, analgesic, used in cough syrups, and parasiticidal.	[34]
Hing	*Ferula assa-foetida*	Ferulic ester		[21]
Oregano	*Origanum vulgare*	Carvacrol and thymol	Tonic, stomachic, used as a water pill, diaphoretic, and emmenagogue.	[23]
Star Anise	*Illicium verum*	Shikimic acid	Astringent, carminative, deodorant, expectorant, and digestive.	[19]
Turmeric	*Curcuma longa*	Curcumin	Thermogenic, palliative, analgesic, anti-inflammatory, vulnerary, depurative, antiseptic, and used in skin diseases.	[35]

**Table 2 antibiotics-12-00270-t002:** Antimicrobial activity of different extraction of spices; M (methanol); E (ethanol); A (acetone); Aq (aqueous); W (water).

S.no.	name	extraction	pathogen	Zone of Inhibition	Reference
1	*Tamarindus indica*	chloroform	*Aspergillus niger* *Aspergillus flavus*	9.0 ± 0.7 11.0 ± 1.8	[145]
2	*Curcuma longa*	Acetone Methanol Ethanol	*B. cereus*	22 20 17	[1]
3	*Emblica officinalis*	Acetone Methanol Ethanol	*B. cereus* *Serratia* sp. *R. mucilaginosa* *A. flavus* *P. citrinum*	22, 19, 17 19, 14, 15 18, 16, 14 19, 17, 12 20, 18, 13	[1]
4	*Ocimum basilicum*	Essential oil	Activity show on Many fungus		[19]
5	*Foeniculum vulgare*	Essential oil	*Bacillus subtilis* *Aspergillus niger*	29 28	[146]
6	*Allium sativum*	Methanol aqueous	*B. subtilis* *C. albicans*	16,20 12	[107]
7	*Coriandrum sativum*	Essential oil	Activity show on Many fungus		[79]
8	*Asafoetida*	Ethanol Methanol Aqueous	*Aspergillus niger* *C. albicans*	15, 17, 13 16, 20, 11	[21]
9	*Trigonella foenum graecum*	Flucanozole Hexane	*Aspergillus niger* *Aspergillus flavus*	16, 10 21, 9	[147]
10	*Murraya koenigii*	Methanol Ethanol Acetone aqueous	*Aspergillus niger*	20 18 11 0	[84]
11	*Nigella sativa*	Ethanol Methanol water	*Aspergillus niger* *Aspergillus fumigatus*	15, 14, 9 17, 19, 7	[148]
12	*Elettaria cardamomum*	Essential oil			[58]
13	*Star anise*	Essential oil	Activity show on Many fungus		[19]

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
