# Peer review of "Current Understanding of the Molecular Basis of Spices for the Development of Potential Antimicrobial Medicine"

_antibiotics, 2023, doi:10.3390/antibiotics12020270_

Round 1
Reviewer 1 Report
The article is interesting, but the following issues should be completed before publication:
- draw chemical formulas of the main components of individual plant materials,
- the quality of Fi.1, Fig. 2 and Fig.-3 should be improved,
- it would greatly enrich this article if it included photos of the plant materials discussed in it,
- please draw detailed mechanisms of antibacterial and antifungal activity of individual raw materials, indicating which organic compounds contained in the plant material are responsible for such a mechanism,
- we use many methods to obtain active substances from plant materials, from the point of view of using the extract as an antibacterial and antifungal preparation, which of the methods used seems to be the best?
- is it better to use extracts that are a mixture of many biologically active substances, or is it better to use pure substances? If we wanted to use isolated, pure organic compounds, what method should be used to isolate and purify them?
Author Response
The article is interesting, but the following issues should be completed before publication:
- draw chemical formulas of the main components of individual plant materials,
Answer: The chemical formula of some spices has been drawn.
- the quality of Fi.1, Fig. 2, and Fig.-3 should be improved,
- it would greatly enrich this article if it included photos of the plant materials discussed in it,
Answer: The quality of the figures is improved and some photos of plant material are added.
- please draw detailed mechanisms of antibacterial and antifungal activity of individual raw materials, indicating which organic compounds contained in the plant material are responsible for such a mechanism,
Answer: In this review, many spices are discussed so drawing the mechanisms of antibacterial and antifungal activity of individual raw materials is difficult.
- we use many methods to obtain active substances from plant materials, from the point of view of using the extract as an antibacterial and antifungal preparation, which of the methods used seems to be the best?
Answer: decoction and Soxhlet extraction are good and easy methods for plant extraction.
- is it better to use extracts that are a mixture of many biologically active substances, or is it better to use pure substances? If we wanted to use isolated, pure organic compounds, what method should be used to isolate and purify them?
Answer: most plants give good activity with a mixture of many biological substances and Chromatography and distillation are used to isolate pure organic compounds.
Reviewer 2 Report
The manuscript contains scientifically important information, however, there is some suggestion/clarification on the manuscript.
1. The paper should be format given by journal guidelines. The authors did not follow the guidelines given by antibiotics from starting to reference section.
2. Table 1. Active compound and medicinal uses of some spices. Add reference for each Medicinal use
3. English writing is very poor.
4. The entire manuscript may undergo a grammar check.
5. The figures are not clear.
Author Response
The manuscript contains scientifically important information, however, there is some suggestion/clarification on the manuscript.
- The paper should be format given by journal guidelines. The authors did not follow the guidelines given by antibiotics from starting to the reference section.
Answer: All references are added as guidelines given by antibiotics.
- Table 1. Active compound and medicinal uses of some spices. Add reference for each Medicinal use
Answer: References are added.
- English writing is very poor.
Answer: English is improved in the revised manuscript.
- The entire manuscript may undergo a grammar check.
Answer: grammar is improved in the revised manuscript.
- The figures are not clear.
Answer: figures are changed.
Reviewer 3 Report
This paper focuses on summarizing the antimicrobial activity of different spices, which is essential for the development of new antimicrobial agents to overcome antimicrobial resistance in humans. The minor issues that need to be modified in this article are: 1) The antimicrobial mechanisms of various spices molecules need to be further supplemented; 2) the size of zone of inhibition in table 2 should be provided; 3) The chemical structural formulas of the main active ingredients in various spices molecules need to be drawn; 4) It is recommended that the following articles (on bacterial multidrug resistance) be referenced in the introduction section, ACS Appl. Mater. Interfaces 2020, 12, 22479; Carbohydrate Polymers 257 (2021) 117636; Small 2021, 2101495.
Author Response
This paper focuses on summarizing the antimicrobial activity of different spices, which is essential for the development of new antimicrobial agents to overcome antimicrobial resistance in humans. The minor issues that need to be modified in this article are: 1) The antimicrobial mechanisms of various spices molecules need to be further supplemented; 2) the size of zone of inhibition in table 2 should be provided; 3) The chemical structural formulas of the main active ingredients in various spices molecules need to be drawn; 4) It is recommended that the following articles (on bacterial multidrug resistance) be referenced in the introduction section, ACS Appl. Mater. Interfaces 2020, 12, 22479; Carbohydrate Polymers 257 (2021) 117636; Small 2021, 2101495.
Answer: In this review, we discussed many spices and their antimicrobial properties but drawing the mechanism of all spices is difficult in our next review we only discussed the mechanism.
In table 2 size of the zone of inhibition is given.
The chemical formula of some spices is added.
Round 2
Reviewer 1 Report
The article may be published.